

# RNA-seq reveals Nup62 as a potential regulator for cell division after traumatic brain injury in mice hippocampus

Jianwei Zhao[1,2], Weihua Wang[1], Ke Yan[1], Haifeng Zhao[3], Zhen Zhang[1], Yu Wang[1], Wenyu Zhu[1] and Shiwen Chen[2]

[1] Department of Neurosurgery, Suzhou Science & Technology Town Hospital, Suzhou, Jiangsu Province, China
[2] Department of Neurosurgery, Shanghai Sixth People's Hospital Affiliated to Shanghai Jiao Tong University School of Medicine, Shanghai, Shanghai, China
[3] Department of Pathology, Suzhou Science & Technology Town Hospital, Suzhou, Jiangsu Province, China

## ABSTRACT

**Background**. Hippocampus impairment is a common condition encountered in the clinical diagnosis and treatment of traumatic brain injury (TBI). Several studies have investigated this phenomenon. However, its molecular mechanism remains unclear.
**Methods**. In this study, Illumina RNA-seq technology was used to determine the gene expression profile in mice hippocampus after TBI. We then conducted bioinformatics analysis to identify the altered gene expression signatures and mechanisms related to TBI-induced pathology in the hippocampus. Real-time quantitative polymerase chain reaction and western blot were adopted to verify the sequencing results.
**Results**. The controlled cortical impact was adopted as the TBI model. Hippocampal specimens were removed for sequencing. Bioinformatics analysis identified 27 upregulated and 17 downregulated differentially expressed genes (DEGs) in post-TBI mouse models. Potential biological functions of the genes were determined *via* Gene Set Enrichment Analysis (GSEA)-based Gene Ontology (GO) and Kyoto Encyclopedia of Genes and Genomes (KEGG) analyses, which suggested a series of functional changes in the nervous system. Specifically, the nucleoporin 62 (Nup62) DEG was discussed and verified. Gene ontology biological process enriched analysis suggests that the cell division was upregulated significantly. The present study may be helpful for the treatment of impaired hippocampus after TBI in the future.

## INTRODUCTION

Temporary or permanent cognitive impairment is one of the most common complications of traumatic brain injury, mainly manifested as memory loss, poor concentration, and reduced executive ability (*Maas, Stocchetti & Bullock, 2008*; *Paterno, Folweiler & Cohen, 2017*; *Sinke et al., 2021*). The hippocampus is crucial to memory (*Kim et al., 2015*). It is a vital brain region involved in the physiological circuitry of memory and is often damaged after TBI (*Graham et al., 1995*). The destruction of the hippocampus is a pathological feature of the human and animal models of brain injury (*Carbonell & Grady, 1999*; *Graham et al., 1995*). TBI includes primary and secondary injuries, and the hippocampus is highly

Corresponding authors
Wenyu Zhu, zwy2000@sina.com
Shiwen Chen, chenshiwen@126.com

susceptible to TBI (*Ansari, Roberts & Scheff, 2008*). The primary brain injury is caused at the moment of the impact, which involves direct tissue damage, impaired regulation of cerebral blood flow, and diffuse axonal injury due to shearing, tearing, or stretching (*Prasetyo, 2020*). The secondary injury was complex and involved a cascade of ischemia, anoxia, and cytotoxic and inflammatory processes (*Prasetyo, 2020*). After TBI, hippocampal atrophy is a common problem in various experimental models of TBI (*Bramlett & Dietrich, 2002*; *Saber et al., 2017*) that contribute to hippocampal-dependent memory impairments. Changes were also observed in the hippocampal structure at the cellular level. Fourier transform infrared imaging (FTRI) identified TBI-caused significant structural changes with respect to total protein content, lipid content, lipid/protein ratio, and membrane lipid order (*Cakmak et al., 2016*; *Ustaoglu et al., 2021*). The dysregulation of neurotransmitters, including gamma-aminobutyric acid (GABA) and glutamate (*Almeida-Suhett et al., 2015*; *Harris et al., 2012*), led to hippocampal deficits. Also, biochemical compounds and changes in electrical neural activity affected different subsections of the rodent hippocampus following TBI (*Girgis et al., 2016*).

As mentioned above, pathological changes occur in the hippocampus after TBI at all levels (*Almeida-Suhett et al., 2015*; *Bramlett & Dietrich, 2002*; *Cakmak et al., 2016*; *Girgis et al., 2016*; *Harris et al., 2012*; *Saber et al., 2017*; *Ustaoglu et al., 2021*). However, the mechanism of injury and modulation in the hippocampus post-TBI is yet controversial. Our results identified the putative genetic changes and potential therapeutic targets of hippocampal injury in mice after moderate TBI using Illumina RNA-seq technology.

Strikingly, RNA sequencing technology has become a powerful research tool for exploring disease pathogenesis. In previous studies, transcriptome analyses were used to identify gene changes in the hippocampus post-TBI at later time points, such as 3 or 7 days (*Attilio et al., 2021*; *Todd et al., 2021*). As the effect of TBI on hippocampal brain tissue was time-dependent (*Ustaoglu et al., 2021*), in this study, 24 h post-TBI was chosen as the time point. As affected by primary and secondary injuries, the whole hippocampus tissue was examined in this research. We used a systematic approach to identify novel molecular biomarkers. In addition, the differently expressed gene (DEG) profiles were explored based on sequencing data. Functional enrichment analyses determined the hub gene-related functions. The top hub gene Nup62 was chosen, and the upregulation of Nup62 was determined by both real-time quantitative polymerase chain reaction (RT-qPCR) and western blot (WB) assay. These findings could provide a novel insight into TBI-induced hippocampus impairment.

## MATERIALS & METHODS

### Animals and experimental groups

Male C57B6/J 8–10-weeks-old mice (20–25 g; SLAC Laboratory Animal Corporation, Shanghai, China) used in this work were bred and housed (five per cage) in standard cages under a 12-hour light-dark cycle with the controlled temperature of 23 $\pm$ 2 °C and full access to food and water. A total of 24 mice were randomly divided into two groups: TBI ($n = 12$) and sham-operated ($n = 12$). In each group, three mice as biological replicates were used for RNA sequencing, three used for (RT-qPCR), four for WB, and

two for hematoxylin-eosin (HE) staining. All procedures in these studies involving animals followed the protocols approved by Suzhou Institute of Biomedical Engineering and Technology, Chinese Academy of Sciences, Suzhou, China (Approval No: 2021-B28). Samples were collected 24 h after the models were established. All mice were euthanized humanely *via* cervical spondylectomy.

## Establishment of the TBI mouse model

Controlled cortical impact (CCI) was adopted as the TBI model. After the mice were anesthetized with 1% ketamine (75 mg/kg) and xylazine (10 mg/kg), their heads were fixed in a stereotaxic frame (Stoelting, Wood Dale, IL, USA), and their body temperature was kept at 37 °C placing a warming pad under the body. Then, a 10-mm-long incision was made from the midline of the skull under aseptic conditions. After separating the skin and fascia using a vascular clamp, we performed a craniotomy over the center of the right parietal bone, one mm lateral to the sagittal suture using a 4-mm trephine. Mice were excluded from the study if the dura mater was damaged during surgery. The TBI mouse model was established using a CCI device (PinPoint Precision Cortical Impactor PCI3000; Hatteras Instruments Inc., Cary, NC, USA) to simulate a moderate TBI based on our previous studies (*Gong et al., 2022*; *Jing et al., 2020*; *Yuan et al., 2016*). The brain injury of moderate severity was induced using CCI with the following parameters: impact velocity 1.5 m/s, dwell time 100 ms, and striking depth 1.5 mm. The bleeding was staunched with a sterile cotton compress. Bone wax was covered over the surface of the cortex, and the incision was closed carefully with interrupted 6–0 silk sutures under aseptic conditions. The sham group underwent the same procedure except for the impact injury. During the experiment, the body temperature of the mice was stabilized using a heating pad until their consciousness recovered fully; then, all mice were tagged and returned to their cages.

## Hippocampal extraction and mRNA sequencing

The mice were anesthetized and sacrificed 24 h after TBI, and their brains were harvested immediately. The hippocampal tissues were separated from the brains according to Paxinos and Franklin's Mouse Brain Atlas (*Paxinos & Franklin, 2013*). The scalp of the mice was cut to expose the head after they were killed by cervical spondylectomy humanely. The mouse skull was pulled apart with tweezers. Then, the superficial cerebral cortex was carefully removed with straight forceps to expose the underlying hippocampal tissue. Similarly, the contralateral hippocampal tissue was also exposed. Finally, the hippocampus was isolated from the surrounding tissue. The brain was sectioned and stained with HE (see the photograph and HE stained section in the Supplemental Files). Total RNA was extracted using MolPure® Cell/Tissue Total RNA Kit (Yeasen Biotechnology (Shanghai) Co., Ltd., Shanghai, China). After generating cDNA libraries, RNA-seq was performed on the Illumina NovaSeq platform according to the manufacturer's protocol. The RNA-seq datasets have been deposited in the Gene Expression Omnibus (GEO) database under GEO: Message Body GSE214701.

## Bioinformatics analysis

In the current analysis, FastQC (http://www.bioinformatics.babraham.ac.uk/projects/fastqc) method was adopted for quality control. HISAT2 (http://ccb.jhu.edu/software/hisat2) was used to align the raw RNA-seq reads to the mouse reference genome (Mus musculus, GRCm38) and StringTie (http://ccb.jhu.edu/software/stringtie) to assemble and quantify the transcripts. The DEGs between the TBI and sham groups were analyzed using the "DESeq2" R package (version 1.38) in R (version 4.2.2) statistical software (*R Core Team, 2022*) (settings: $P < 0.05$; fold-change (FC)>1.5 or <0.667; false discovery rate (FDR)<0.1). The principal component analysis (PCA) plot was used to determine the principal components using the DEGs list and visualized using the "ggbiplot" R package (version 0.55). The clustering analysis of DEGs was carried out using the "pheatmap" package (version 1.0.12) of R.

The GSEA method with all genes was used for gene enrichment analysis. Functional terms were retrieved from the GO database that describes the gene attributes, including the biological process (BP), molecular function (MF), and cellular component (CC). Then, the analysis was performed to identify significantly different regulatory pathways using the Kyoto Encyclopedia of Genes and Genomes (KEGG), a major public pathway-related database. GSEA-based GO and KEGG analyses were performed using the "clusterProfiler" package (version 4.4.4; https://bioconductor.org/packages/release/bioc/html/clusterProfiler.html) of R statistical software.

## RT-qPCR

For RT-qPCR, total RNA was isolated using the MolPure® Cell/Tissue Total RNA Kit (Yeasen Biotechnology) and reverse transcribed into cDNA. qPCR was carried out using gene-specific primers and $\alpha$-tubulin as the reference gene. Gene-specific primers used for amplification are listed in the Supplemental Files. Then, the mRNA relative expression levels were analyzed *via* RT-qPCR on a LightCycler 480 (Roche).

## Western Blot assays

The isolated hippocampal tissue was homogenized in lysis buffer by ultrasonication, diluted in loading buffer (Cat# P0015; Beyotime) to estimate the protein concentration, and separated by SDS-PAGE. Subsequently, the protein was transferred to the PVDF membrane. The membranes were blocked for 1.5 h with 5% non-fat milk and probed with primary antibodies (Nup62; Proteintech, Rosemont, IL, USA, 1:10000) and $\alpha$-tubulin (Proteintech, 1:15000)) overnight at 4 °C. The membrane was incubated with horseradish peroxidase (HRP)-labeled goat anti-mouse secondary antibody for one hour at room temperature, and the immunoreactive bands were visualized using Amersham Imager 680. The relative optical density was calculated by ImageJ software (NIH, USA).

## Statistical analysis

Continuous variables were presented as means ± SD, and categorical variables were displayed as a percentage. The statistical difference between TBI and sham-operated groups was analyzed using the two-tailed unpaired $t$-test. GraphPad Prism 9 (GraphPad Software, San Diego, CA, USA) was used for analyses and graphic representation of data. $P$-values <0.05 were considered statistically significant.

## RESULTS

### Quality control and assembly of the raw sequence reads

At 24 h after surgery, both the sham and TBI animals were sacrificed, and hippocampus samples were removed rapidly. The size of all hippocampus samples was >0.5 mg, and the 28s/18s value was >0.7 on the Illumina NovaSeq 6000 sequencer used for sequencing. A total of 53.06 million, 49.11 million, 48.80 million, 53.32 million, 40.73 million, and 44.63 million reads were obtained from Sham1, Sham2, Sham3, TBI1, TBI2, and TBI3, respectively, of which 97.63% (Sham1), 97.67% (Sham2), 97.64% (Sham3), 97.61% (TBI1), 97.40% (TBI2), and 97.55% (TBI3) were aligned to the mouse reference genome using HISAT (Table 1). After the reads were compared to the mouse genome, StringTie was used to annotate and quantify the expression (*Pertea et al., 2016*).

### Screening of DEGs

The statistical power of our RNA-seq data calculated by "RNASeqpower" package was 0.8601569. A total of 44 genes were identified to be differentially expressed between TBI and sham groups using the "DESeq2" package of R; of these, 27 were upregulated and 17 were downregulated (settings: $P < 0.05$; fold-change (FC) < 1.5 or < 0.667; FDR < 0.1). Nup62 was the most differentially upregulated gene. Pcdhgb8 was the most differentially downregulated gene. The normalized list of all gene data is available in the Supplemental Files. A PCA plot (Fig. 1) was drawn to assess the variability of the data. The PC1 and PC2 explained 36.3% and 32.6% of the variance in the data, respectively, and the TBI group was isolated from the sham group. The top five most significantly up/downregulated DEGs between TBI and sham groups are listed in Table 2. The volcano and heat maps of all the DEGs are depicted in Fig. 2. Top five significant DEGs identified were labeled in the volcano map.

### Functional enrichment and pathway analyses of DEGs

In our study, the GSEA method was used for gene enrichment analysis. The list of genes used in GSEA was ranked according to $\log_2 FC$. Then GSEA was conducted with two gene set collections: GO and KEGG. The enrichment dot plot of GO was used to describe the BP, CC, and MF entries of all the genes between the TBI and sham groups. Figure 3 illustrates the top 10 upregulated GO terms, including "intermediate filament-based process", "keratin filament" and "keratin filament binding", the most significantly enriched GO terms belonging to BP, CC, and MF, respectively. BP refers to a series of events accomplished by one or more ordered assemblies of MFs. In this study, we focused on the BP of Nup62. Table 3 lists the top eight significantly enriched functions that Nup62 is involved in the GO term of BP. "Positive regulation of cytokinesis", "regulation of cytokinesis" and "positive regulation of centriole replication" were most significantly enriched in the BP group. After the KEGG pathway enrichment analysis, Fig. 4 shows the top 10 most enriched KEGG pathways of activated and suppressed enrichment hallmarks terms, respectively. These included "Glyoxylate and dicarboxylate metabolism", "Renin-angiotensin system", "Cytosolic DNA-sensing pathway" and "Olfactory transduction."

**Table 1  Summary of sequencing and mapping results.**

| Sample | Total_mapped | Unmapped | Ratio |
|---|---|---|---|
| Sham1 | 53057380 | 1287942 | 97.63% |
| Sham2 | 49114576 | 1170052 | 97.67% |
| Sham3 | 48798176 | 1179246 | 97.64% |
| TBI1 | 53325863 | 1306151 | 97.61% |
| TBI2 | 40726291 | 1088909 | 97.40% |
| TBI3 | 44632092 | 1122300 | 97.55% |

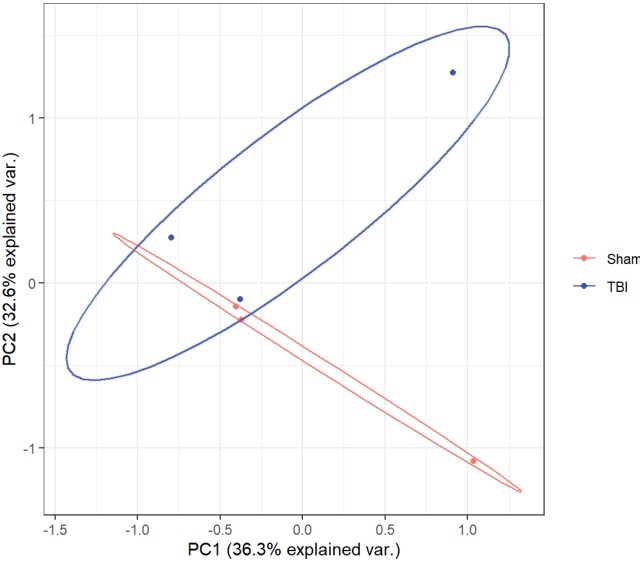

**Figure 1  PCA plot for all DEGs between TBI *versus* Sham.**  TBI, TBI group, marked in blue; Sham, Sham-operated group, marked in red.

**Table 2  Top five most significantly up/downregulated DEGs between TBI and sham groups.**

| Gene symbol | P value | Padj | Status |
|---|---|---|---|
| Nup62 | 1.49E−25 | 1.72E−21 | Up |
| Ldah | 6.04E−11 | 4.64E−07 | Up |
| Klk6 | 1.83E−09 | 8.42E−06 | Up |
| Apod | 3.97E−09 | 1.52E−05 | Up |
| Snx10 | 6.38E−08 | 0.00021 | Up |
| Pcdhgb8 | 9.30E−30 | 2.14E−25 | Down |
| Tmem70 | 1.05E−09 | 6.05E−06 | Down |
| Ptgs2 | 6.76E−07 | 0.001794 | Down |
| Alkal2 | 1.33E−06 | 0.002785 | Down |
| 6430548M08Rik | 1.66E−05 | 0.023855 | Down |

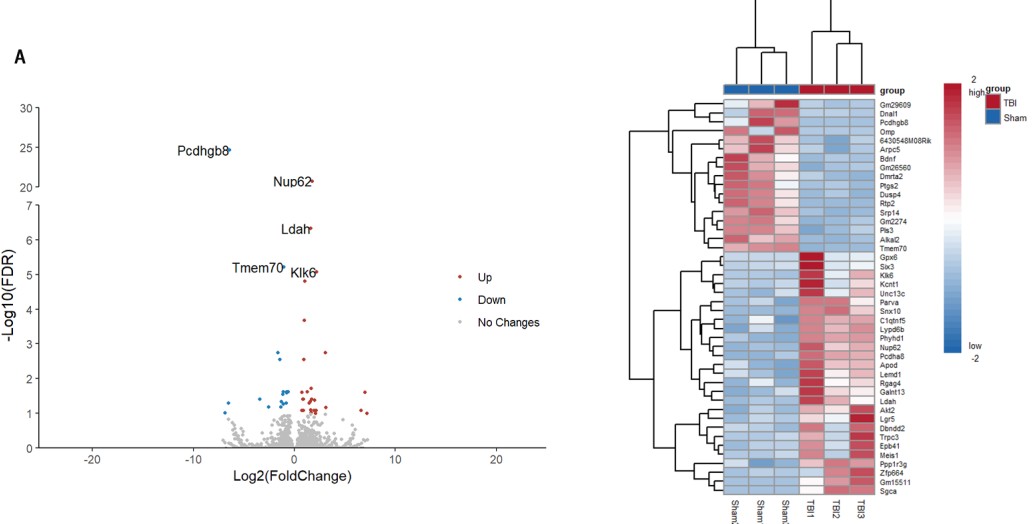

**Figure 2** **(A) Volcano plot of the DEGs. (B) Heatmap for all DEGs between TBI *vs.* sham samples.** (A) Red, upregulation; blue, downregulation. (B) Red, upregulation; blue, downregulation.

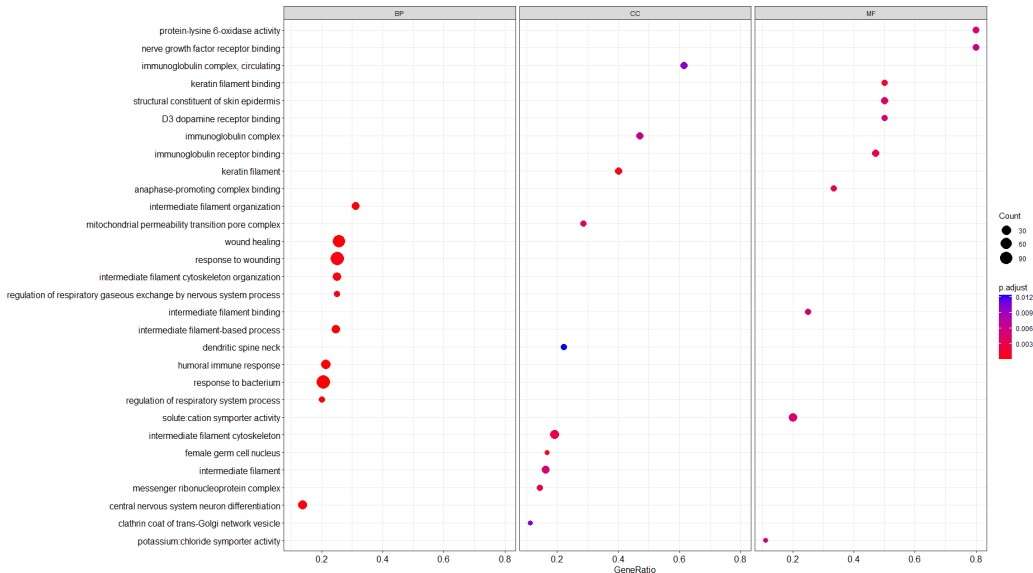

**Figure 3** **Top 10 most significantly enriched GSEA-based GO terms in the three functional groups (compared to the sham).** MF, molecular function; CC, cellular component; BP, biological process. The intensity of the color depends on the Padj value. The size of plot depends on the gene count.

## Nup62 was increased in TBI mice

Biochemical experiments verified the results (Fig. 5). The relative mRNA expression level measured by RT-qPCR showed significantly upregulated Nup62 in the TBI groups compared to the sham groups (Fig. 5A, $P < 0.05$). In WB assays, Nup62 also increased considerably in the TBI groups (Fig. 5B, $P < 0.01$).

**Table 3   Top eight significant functions with respect to Nup62 involved in the GO term of biological process.**

| goID | goDescription | enrichmentScore | *P* value |
|---|---|---|---|
| GO:0032467 | Positive regulation of cytokinesis | 0.656413259 | 0.027891 |
| GO:0032465 | Regulation of cytokinesis | 0.510884018 | 0.044049 |
| GO:0046601 | Positive regulation of centriole replication | 0.770285358 | 0.078899 |
| GO:0032954 | Regulation of cytokinetic process | 0.904279421 | 0.081439 |
| GO:0046599 | Regulation of centriole replication | 0.617102394 | 0.095986 |
| GO:0000910 | Cytokinesis | 0.392821842 | 0.122314 |
| GO:0007099 | Centriole replication | 0.475303426 | 0.220104 |
| GO:0051781 | Positive regulation of cell division | 0.414488468 | 0.247292 |

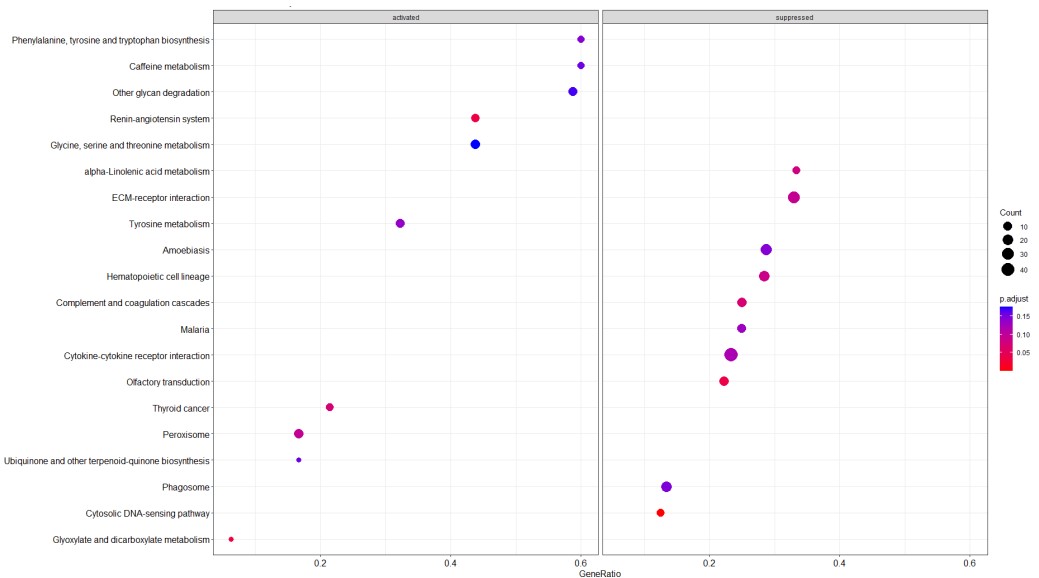

**Figure 4   Top 20 GSEA-based KEGG pathways.** The intensity of the color depends on the Padj value. The size of plot depends on the gene count.

## DISCUSSION

In this study, we performed a bioinformatic analysis of high-throughput sequencing to determine the gene expression profiles of mice in the hippocampus 24 h after TBI. The results showed critical roles for several genes and pathways in the hippocampus pathology after TBI.

A total of 27 up- and 17 downregulated DEGs were detected in the hippocampus of TBI mice. Several pathways enriched for TBI were identified. The GO analysis demonstrated that the "intermediate filament-based process", "keratin filament" and "keratin filament binding" were the most significantly enriched GO terms containing the ranked genes belonging to BP, CC, and MF, respectively. After TBI, a series of functional changes occur in the nervous system, including "central nervous system neuron differentiation"

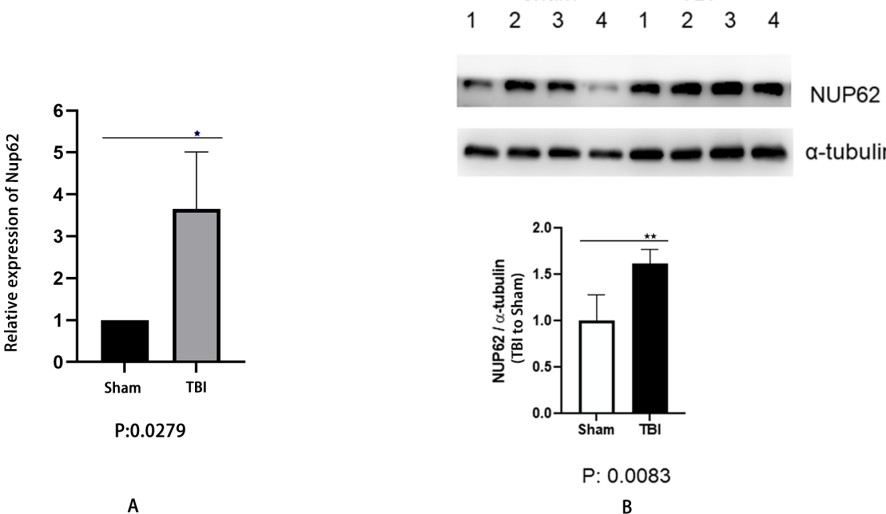

**Figure 5** **RT-qPCR and WB result in TBI mice vs. sham mice.** (A) Relative mRNA expression level measured by RT-qPCR showed that Nup62 was upregulated in the TBI group. (B) WB analysis showed a significant increase in NUP62 protein expression in the TBI group. **P < 0.01, *P < 0.05.

and "nerve growth factor receptor binding". The KEGG pathway analysis revealed that "Glyoxylate and dicarboxylate metabolism" and "Cytosolic DNA-sensing pathway" were the most enriched pathways. These bioinformatics analyses may improve the understanding of hippocampal pathological processes in TBI, especially in the acute phase.

According to the DEGs analysis, Nup62 levels differed significantly between the TBI and sham-operated groups. These findings were validated by RT-qPCR and WB analyses. Therefore, we were concerned about the role of Nup62 in hippocampal pathology after TBI.

Nup62, located on human chromosome 19, is a component of the nuclear pore complex (NPC) and is expressed in various human tissues (*Fagerberg et al., 2014*). NPC is the only channel responsible for material transport in the nuclear membrane that is conserved across all eukaryotes (*Huang et al., 2020*). It also plays a critical role in regulating gene expression, and its abnormal function gives rise to a variety of diseases (*Huang et al., 2022*; *Huang et al., 2020*). Recent studies have investigated the fine structure of NPCs using a comprehensive technique (*Allegretti et al., 2020*; *Beck & Hurt, 2017*; *Fontana et al., 2022*; *Tai et al., 2022*). The molecular mass of vertebrate NPCs ranges from 110–125 MDa and the diameter is about 120 nm (*Fontana et al., 2022*). The NPCs are divided into four main rings: the cytoplasmic ring (CR) on the cytoplasmic side, the inner ring (IR) and luminal ring (LR) on the nuclear membrane plane, and the nuclear ring (NR) facing the nucleus (*Fontana et al., 2022*). The CR provides docking sites for cytoplasmic filaments. It consists of the Nup214 complex formed with Nup62 together with Nup214 and Nup88 (*Tai et al., 2022*; *Wang et al., 2016*; *Wu et al., 2016*). Nup62 plays a critical role in nuclear transportation, cell migration, and cell cycle regulation (*Wu et al., 2016*). A previous study showed that the neuropathology of chronic traumatic encephalopathy

(CTE) patients is associated with the upregulation of the Nup62 gene (*Anderson et al., 2021*). Increased Nup62 *in vivo* and *in vitro* may trigger TDP-43 cytoplasmic and nuclear mislocalization, abnormalities, perinuclear accumulation, and reduced motor ability and lifespan of animals. Thus, modulating Nups, including Nup62 post-trauma, exerted a protective effect following head trauma (*Anderson et al., 2021*). In the chronic stress model, the altered Nup62 levels may affect the architecture and plasticity of apical dendrites in the hippocampus (*Kinoshita et al., 2014*). Another study reported that depletion and cytoplasmic mislocalization of Nup62 contributes to TDP-43 proteinopathy in amyotrophic lateral sclerosis (ALS)/frontotemporal lobar degeneration (FTLD) (*Gleixner et al., 2022*). Nup62 is also associated with other neurodegenerative diseases, such as Alzheimer's and Huntington's disease (*Nag & Tripathi, 2022*). In conclusion, Nup62 pathology may be a common event in various nervous system disorders. Thus, we speculated that changes in Nup62 are associated with a poor prognosis of TBI.

Furthermore, our data in GO analysis suggested that altered Nup62 may cause changes in the cell cycle that lead to TBI pathology. In the GO_BP group, "positive regulation of cytokinesis", "regulation of cytokinesis", "positive regulation of centriole replication" and other cell division-related pathways were enriched post-TBI compared to the sham group. Cell division depends on the activation of the cell cycle. After TBI, cell cycle activation (CCA) occurs in the hippocampus cells, including neurons, glial cells, and progenitor cells, and contributes to secondary brain injury (*Redell et al., 2020*; *Stoica, Byrnes & Faden, 2009*). In the animal models of TBI, CCA in the brain has been well-demonstrated experimentally (*Kabadi et al., 2012a*; *Kabadi et al., 2014*; *Skovira et al., 2016*). Reportedly, cell cycle proteins are upregulated in post-mitotic cells, including neurons and mature oligodendrocytes, and proliferating cells, including microglia and astrocytes (*Skovira et al., 2016*). In the proliferating cells, CCA induced the formation of glial scar and produced the neuroinflammatory factor that ultimately led to neuronal degeneration (*Kabadi et al., 2012b*; *Loane & Byrnes, 2010*; *Stoica, Byrnes & Faden, 2009*). For post-mitotic cells, re-entry into the cell cycle is associated with apoptotic cell death (*Skovira et al., 2016*). Specifically, enhanced neurogenesis and increased proliferation of progenitor cells are observed in the hippocampus after TBI (*Liu & Song, 2016*). The magnitude of injury is correlated with the degree of post-TBI neurogenesis in the hippocampus (*Girgis et al., 2016*; *Wang et al., 2016*). Abnormal neurogenesis in the hippocampus may result in detrimental effects, including aberrant sprouting and migration, reduced dendritic outgrowth, and loss of newborn neurons (*Gibb et al., 2015*; *Ibrahim et al., 2016*; *Robinson, Apgar & Shapiro, 2016*). Taken together, CCA has an adverse impact on hippocampal function. Moreover, the cell cycle inhibitors improve the functional outcomes following TBI in several models (*Kabadi et al., 2012b*; *Kabadi et al., 2012c*; *Kabadi et al., 2014*). Previous studies proved that NPCs, including Nup62, regulate the gene expression at the NPC and within the nucleoplasm (*Kalverda et al., 2010*). Nucleoporin-chromatin interactions stimulate the cell-cycle gene expression directly inside the nucleoplasm (*Casolari et al., 2004*; *Kalverda et al., 2010*; *Taddei et al., 2006*). Therefore, we speculated that Nup62 affects the hippocampal function after TBI by activating the cell cycle. Nonetheless, additional *in vivo* and *in vitro* studies are required to verify this finding further.

Since the present study only involves the changes in the acute phase after TBI, we observed the alteration 24 h after TBI. One limitation of the present study was the lack of experimental ethology. The experiments, such as Morris water maze and T-maze, are required to evaluate hippocampus impairment, which leads to cognitive deficits, memory difficulties, and behavioral disorders. In addition, further study is needed to address whether Nup62 dysfunction is a cause or a consequence of the hippocampus pathology in TBI to understand the exact mechanism. The correlation between the specific CCA mechanism and central nervous system damage also needs to be explored at the molecular level.

## CONCLUSIONS

The bioinformatics analysis of DEGs showed that Nup62 mRNA was significantly upregulated in the acute stage. The biochemical experiments confirmed this conclusion at the RNA and protein levels. Post-trauma, the Nup62 protein may be upregulated at the transcriptional level. The GO_BP enrichment analysis showed that the cell division of mice after TBI treatment was significantly elevated. The data from our experiments suggest that Nup62 enhances cell division in TBI mice. Further experimental investigations on cell division after TBI should be considered. Also, the long-term effect of Nup62 after TBI needs further investigation.

## ACKNOWLEDGEMENTS

We thank our colleagues, especially Qiuyuan Gong, for their valuable assistance in the experiment.

### Funding

This work was supported by the project of Shanghai Science and Technology Commission (No.19ZR1438600), the Science and Technology Development Fund of Nanjing Medical University (No.NMUB20210251), and the Pre-Research Fund of Suzhou Science & Technology Town Hospital (No.szkjcyy2021004). The funders had no role in study design, data collection and analysis, decision to publish, or preparation of the manuscript.

### Grant Disclosures

The following grant information was disclosed by the authors:
The project of Shanghai Science and Technology Commission: No. 19ZR1438600.
Science and Technology Development Fund of Nanjing Medical University: No. NMUB20210251.
Pre-Research Fund of Suzhou Science & Technology Town Hospital: No. szkjcyy2021004.

### Competing Interests

The authors declare there are no competing interests.

## Author Contributions

- Jianwei Zhao conceived and designed the experiments, performed the experiments, analyzed the data, authored or reviewed drafts of the article, and approved the final draft.
- Weihua Wang performed the experiments, prepared figures and/or tables, and approved the final draft.
- Ke Yan analyzed the data, prepared figures and/or tables, and approved the final draft.
- Haifeng Zhao performed the experiments, prepared figures and/or tables, and approved the final draft.
- Zhen Zhang analyzed the data, prepared figures and/or tables, and approved the final draft.
- Yu Wang analyzed the data, authored or reviewed drafts of the article, and approved the final draft.
- Wenyu Zhu conceived and designed the experiments, authored or reviewed drafts of the article, and approved the final draft.
- Shiwen Chen conceived and designed the experiments, authored or reviewed drafts of the article, and approved the final draft.

## Animal Ethics

The following information was supplied relating to ethical approvals (i.e., approving body and any reference numbers):

Suzhou Institute of Biomedical Engineering and Technology, Chinese Academy of Sciences, Suzhou, China (Approval No: 2021-B28) provided full approval for this research.

## Data Availability

The data is available at NCBI GEO: GSE214701.

## Supplemental Information

Supplemental information for this article can be found online at http://dx.doi.org/10.7717/peerj.14913#supplemental-information.

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
