# Peer review of "RNA-seq reveals Nup62 as a potential regulator for cell division after traumatic brain injury in mice hippocampus"

_PeerJ, doi:10.7717/peerj.14913_

## Round 0.1 · original submission · Major Revisions

Please answer the reviewer's comments carefully.

Reviewer 1 ·

Basic reporting

This manuscript utilizes RNA-seq assay to determine the differentially expressed genes (DEGs) in the hippocampus after traumatic brain injury (TBI). The authors identified Nup62 as the most significantly upregulated DEG and verified its expression after TBI. This study may provide more potential targets for the treatment of impaired hippocampus after TBI. Overall, the manuscript is well-organized and clear. However, there are parts in this manuscript that can be further improved.

-line 60-61. ‘Physiological changes were linked to memory’. Please explain more the reason why the authors mentioned ‘physiological changes’, I didn’t see any connections in the context. Also, please provide citations.

-line 118: Is it (Fig. 1B) or (Fig. 2)? Please check all the other figure labelling in the main text.

Experimental design

-line 65-66: ‘bilateral hippocampus tissues were examined in this research’. However, I didn’t see any results of the comparison between the left and right hippocampus in experimental and control groups.

-line 79: How about technical replicates? Did the authors pool the samples together?

-line 83 and 268: the authors collected the samples after 24h, will the time point well represent the first and secondary injuries? Are there any other studies using the same time point.

Validity of the findings

-The authors should provide the PC plot for the RNA-seq analysis. Additionally, please provide the volcano plot for the DEGs, and label the top 5 or 10 genes identified in this study, which will be much better for audience to visualize, instead of looking at the excel data.

Reviewer 2 ·

Basic reporting

This paper need significantly improvement on the grammar and writing. Additionally, for methods part, from lines 83-90, it is not necessary to be included in the method as long as the animal protocols are approved with the institution. The results part are too brief, the authors may want to extend the writing of this part.

Experimental design

This study aims to profile transcriptome in the hippocampus to understand the molecular mechanism underlying brain traumatic injury (TBI), which is an important question, and certainly within the aim and scope of this journal.
However, the authors failed to introduce sevevral previous transcriptome study related to the TBI (Todd, Brittany P., et al. "Traumatic brain injury results in unique microglial and astrocyte transcriptomes enriched for type I interferon response." Journal of neuroinflammation 18.1 (2021): 1-15.
Attilio, Peter J., et al. "Transcriptomic analysis of mouse brain after traumatic brain injury reveals that the angiotensin receptor blocker candesartan acts through novel pathways." Frontiers in Neuroscience 15 (2021): 636259.), and how different is their model and approaches comparing to previous works.

Validity of the findings

The authors should validate their TBI procedure with either known phenotypic or molecular markers. Otherwise, it is hard to judge whether the TBI is established correctly or not. Figure 1 can be omitted.
For Figure 2, the authors might want to show the staining of all their TBI and sham mice to show consistent results.

As for the bioinformatics analyses of RNA-Seq, it was not clear how authors performed DEG analyses, and whether DEGs were filtered properly with FDR cutoff. From their methods, it seems that only P-value was used rather than FDR. In RNA-Seq data analyses, multiple testing correction (FDR) is crucial to identify true DEGs. And based on the heatmap of figure 3, the DEGs expression seems to be very variable between TBI samples, thus might likely be false positive DEGs. The authors may want to use PCA analysis to show that there is distinct transcriptome differences between TBI and sham groups.

For figure 4 the GO term analyses, the labeling of the x-axis is missing, and the legend is not clear enough.

For figure 6, statistical testing of RT-qPCR was not performed.

The GEO accession was not public and the token was not provided by the authors. The reviewer cannot access the raw data to further evaluate the data quality.

Reviewer 3 ·

Basic reporting

1.1. The figures 1 and 5, which have parts A and B, must be uploaded as a single figure, i.e., a single figure 1 (with parts A and B), and a single figure 5 (with parts A and B). This may be better to review and to configure figure format.
1.2. Figure 4 in PDF's format (10 most significantly enriched GO terms): please, improve the figure quality.
1.3. Figure 7 in PDF's format (RT-qPCR and WB result in TBI mice vs. sham mice): Please, divide it into parts A (for the first result) and part B (for the Norm to Sharm result). On part B, it is not clear if you are applying the hypothesis test of Sham to the TBI group, if so, I suggest to put the ** above a line above the both rectangles.
1.4. Table 1: please, remove the underline on "total_mapped".
1.5. Table 2: must include adjusted P-value (padj).
1.6. In supplementary file wb_expression.xlsx, you have columns D and E not specified with a header, please specify or remove it.

Experimental design

No comment.

Validity of the findings

The differentially expressed analysis was carried out by analyzing data from the settings: P<0.05; fold-change (FC)>1.5 or <0.667.
3.1. Why to choose FC, or log2FC if that is it, > 1.5 or <0.667? What criteria was used for that?
3.2. We have a problem in the analysis, the P-value resulted in DESeq2 package is just an illustrative value, the statistical result that should be considered is the Adjsted P-Value (padj column) to analyze all the results. Here you can check why this analysis is inappropriate: https://www.ncbi.nlm.nih.gov/pmc/articles/PMC6099145/
Please, consider to change the analysis to focus on the padj, you can report it at the methodology topic as the following authors did:
a) https://www.sciencedirect.com/science/article/pii/S1095643322000277
b) https://www.frontiersin.org/articles/10.3389/fimmu.2021.661437/full
c) https://www.ncbi.nlm.nih.gov/pmc/articles/PMC8593470/pdf/fgene-12-730991.pdf

Additional comments

4.1. I see that the word "mice" must be included in title. For example: "... brain injury in mice hippocampus". The same for line 24.
4.2. LINE 33: change GO_BP to gene ontology biological process.
4.3. LINE 34: change "showed" to "suggests", once the GO analysis suggests to you, you did not prove it experimentally with others methods and technologies.
4.4. LINE 123: you must report the version for the following packages: DESeq2, pheatmap, cluster profiler. And you must also report the R version.
4.5. LINE 154: you say: "All data...", which data? Please specify.
4.6. LINEs 182:187, 202-203 and 241-242: change the ," for ",
4.7. LINE 204: you say: "After TBI, a series of functional changes occur in the nervous system, including learning or memory, which is consistent with what is observed clinically.". Did you analyzed that?
4.8. LINE 227: there is a space in "(CTE )". Consider change to "(CTE)".
4.9. LINE 283: you say: "The data of our experiments indicated that Nup62 enhances...". Did you prove it experimentally? I recommend change the "indicated" word to "suggests".
4.10: LINES 283 and 284: you did not proved experimentally that it increased cell division. I recommend to change the word "indicated" to "suggested" and include that further experimental investigations must be considered.

---

## Round 0.2 · accepted · Accept

In this revised version. the authors have addressed the questions well. Thanks.

Reviewer 2 ·

Basic reporting

NA

Experimental design

NA

Validity of the findings

NA

Additional comments

The paper has been improved and addressed my concerns

Reviewer 3 ·

Basic reporting

The authors carried out the suggested modifications.

Experimental design

The authors carried out the suggested modifications.

Validity of the findings

The authors carried out the suggested modifications.